# Organic–Inorganic Ternary Nanohybrids of Single-Walled Carbon Nanohorns for Room Temperature Chemiresistive Ethanol Detection

**DOI:** 10.3390/nano10122552

**Published:** 2020-12-18

**Authors:** Cornel Cobianu, Bogdan-Catalin Serban, Niculae Dumbravescu, Octavian Buiu, Viorel Avramescu, Cristina Pachiu, Bogdan Bita, Marius Bumbac, Cristina-Mihaela Nicolescu, Cosmin Cobianu

**Affiliations:** 1National Institute for Research and Development in Microtechnologies–IMT Bucharest, 126 A Erou Iancu Nicolae Str., 077190 Voluntari, Romania; bogdan.serban@imt.ro (B.-C.S.); niculae.dumbravescu@imt.ro (N.D.); octavian.buiu@imt.ro (O.B.); viorel.avramescu@imt.ro (V.A.); cristina.pachiu@imt.ro (C.P.); bogdan.bita@inflpr.ro (B.B.); 2Science and Technology Section, Academy of Romanian Scientists, 3 Ilfov Str., Sector 5, 050045 Bucharest, Romania; 3Sciences and Advanced Technologies Department, Faculty of Sciences and Arts, Valahia University of Targoviste, 13 Sinaia Alley, 130004 Targoviste, Romania; marius.bumbac@valahia.ro; 4Institute of Multidisciplinary Research for Science Technology, Valahia University of Targoviste, 13 Sinaia Alley, 130004 Targoviste, Romania; cristina.nicolescu@valahia.ro; 5Electrical Engineering, Electronics and Information Technology Faculty, Valahia University of Targoviste, 13 Sinaia Alley, 130004 Targoviste, Romania; cosmin.cobianu@valahia.ro

**Keywords:** oxidized single-walled carbon nanohorn (ox-SWCNH), organic–inorganic ternary nanohybrids, SnO_2_ and ZnO nanoparticles, polyvinylpyrrolidone (PVP), electrical percolation threshold, ethanol vapor detection, *p*-type semiconductor, chemiresistive principle, HSAB theory

## Abstract

Organic–inorganic ternary nanohybrids consisting of oxidized-single walled carbon nanohorns-SnO_2_-polyvinylpyrrolidone (ox-SWCNH/SnO_2_/PVP) with stoichiometry 1/1/1 and 2/1/1 and ox-SWCNH/ZnO/PVP = 5/2/1 and 5/3/2 (all mass ratios) were synthesized and characterized as sensing films of chemiresistive test structures for ethanol vapor detection in dry air, in the range from 0 up to 50 mg/L. All the sensing films had an ox-SWCNH concentration in the range of 33.3–62.5 wt%. A comparison between the transfer functions and the response and recovery times of these sensing devices has shown that the structures with ox-SWCNH/SnO_2_/PVP = 1/1/1 have the highest relative sensitivities of 0.0022 (mg/L)^−1^, while the devices with ox-SWCNH/SnO_2_/PVP = 2/1/1 have the lowest response time (15 s) and recovery time (50 s) for a room temperature operation, proving the key role of carbonic material in shaping the static and dynamic performance of the sensor. These response and recovery times are lower than those of “heated” commercial sensors. The sensing mechanism is explained in terms of the overall response of a p-type semiconductor, where ox-SWCNH percolated between electrodes of the sensor, shunting the heterojunctions made between n-type SnO_2_ or ZnO and p-type ox-SWCNH. The hard–soft acid–base (HSAB) principle supports this mechanism. The low power consumption of these devices, below 2 mW, and the sensing performances at room temperature may open new avenues towards ethanol sensors for passive samplers of environment monitoring, alcohol test portable instruments and wireless network sensors for Internet of Things applications.

## 1. Introduction

Considering the safety requirements in chemical and biomedical applications [1,2,3], an intensive research study was performed for the improvement of the static and dynamic behavior of ethanol vapor sensors prepared by surface acoustic wave [4,5,6], optical fiber [7,8,9] and chemiresistive metal oxide semiconductor (MOS) technology [10,11,12,13,14,15]. To prove this, one can follow the continuous development of novel nanocomposites between metal oxides and the carbon allotrope family (single and multi-walled carbon nanotubes (SWCNT and MWCNT)), graphene and single-walled carbon nanohorns (SWCNH) [14,15,16,17].

Following the methodology developed for the functional characterization of the SW and MWCNT, the modification of the electrical properties of the SWCNH in the presence of ambient gas adsorbed on its surface was one of the emerging research topics [18,19,20]. Thus, single-walled carbon nanohorns (SWCNH) and oxidized SWCNH (ox-SWCNH) as single constituents were used as sensing layers in the design of chemiresistive sensors for the detection of reducing gas, ammonia [18], oxidizing gas, nitrogen dioxide [19], and more recently for relative humidity (RH) sensing, as proven in our previous work [20].

These pioneering functional studies of the novel nano-carbonic material were directly focused on the intrinsic capabilities of SWCNHs to interact with ambient gases, in terms of adsorption and surface charge transfer reactions, which took place when this material was applied directly between two metal electrodes. Thus, the *p*-type behavior of SWCNW in the presence of reducing (NH_3_) and oxidizing gas (NO_2_) has been already demonstrated [18,19]. Our recent work proved a similar *p*-type response of the ox-SWCNH for the case of relative humidity (RH) detection in chemiresistive sensing structures with sensing films made of ox-SWCNH [20].

This useful early-stage research has opened the way towards the realization of more elaborate SWCNH-based sensing films, where the initial challenges of rather poor adherence of a pure SWCNH layer placed on a substrate and the limited reliability of such devices needed to be addressed. For this purpose, in our recent paper [21], binary SWCNH-hydrophilic organic polymer nanocomposites used as sensing films for chemiresistive RH sensors were proven to solve these fabrication issues, providing either a switch characteristic (for SWCNH-triblock copolymer) or a linear characteristic (SWCNH-polyvinylpyrrolidone (PVP)) [21,22]. The high concentration of the SWCNHs in the above binary nanocomposites (>33 wt%), which was much higher than the percolation threshold of carbonaceous materials in polymers (around 1 wt%) [22], had a major effect on the electrical conduction process and sensor response. The increase in the electrical resistance of these binary composites as a function of the RH value indicated the p-type behavior and this result was due to the ox-SWCNH percolating the film [23].

Moreover, in the same evolutional direction, a linear RH response of the chemiresistive sensors with films based on ternary nanocomposites consisting of ox-SWCNH-graphene oxide (GO)-PVP with the mass ratio of the three components equal to 1/1/1 was obtained recently in our group [24]. The response time and sensitivity of such a device were quite similar to the ones provided by a commercial capacitive RH sensor, made by a complementary metal oxide semiconductor (CMOS) technology and benefiting from an associated complex signal processing unit.

Organic–inorganic ternary nanocomposites of the SWCNHs consisting of carbonaceous material, semiconducting metal oxide (SnO_2_, ZnO), and binding polymer (PVP) are now emerging, as a step forward in the material research, aiming for further increase in the reliability of the chemiresistive sensors, due to the presence of the ceramic material and its contribution to the sensing mechanism [25,26,27].

It is the purpose of this paper to explore the room temperature ethanol sensing properties of these ternary organic–inorganic nanohybrids, with the motivation of identifying new pathways for the reduction in electric power consumption in next-generation sensors for the Internet of Things applications, where the energetic constraints are prohibiting the use of existing ethanol sensors for portable applications. The focus of this work will be to evaluate the sensing properties of the ternary inorganic–organic nanocomposites of oxidized-SWCNHs in the domain of low concentrations of ethanol in air, as required by mobile instrumentation and alcohol testers.

## 2. Materials and Methods

### 2.1. Materials

Powder of oxidized SWCNHs (ox-SWCNH) (diameter of 2–5 nm, length of 40–50 nm, the specific surface area of 1300–1400 m^2^/g, purity equal to 90% (10% amorphous graphite)), polyvinylpyrrolidone ((PVP), special grade analytical, average mol wt. 40,000), SnO_2_ nanopowder (averaged particle size lower than 100 nm), hexa-hydrated zinc nitrate ((Zn(NO_3_)_2_·6H_2_O), with purity higher than 99%), sodium hydroxide powder (reagent grade, 97%) and isopropyl alcohol (>99.7% FCC, FG) were all purchased from Sigma-Aldrich (Taufkirchen City, Germany). According to the supplier, the hydrophilic ox-SWCNH may form spherical aggregates with a diameter of 100 nm, containing thousands of individual nanohorns. Thanks to the fabrication method, the ox-SWCNHs have no metal contamination and present good dispersibility.

### 2.2. Design of Experiments for Organic–Inorganic Ternary Nanohybrids of Ox-SWCNH Fabrication

For the study of the sensing properties of the ternary organic–inorganic nanohybrids of the ox-SWCNHs, the following chemical compositions of the sensing films were designed and performed, as described below.

Ox-SWCNH/SnO_2_/PVP = 1/1/1, Ox-SWCNH/SnO_2_/PVP = 2/1/1, Ox-SWCNH/ZnO/PVP = 5/3/2, and Ox-SWCNH/ZnO/PVP = 5/2/1, all mass ratios (*w*/*w*/*w*).

SnO_2_ nanopowder, as described above, was used for the first two compositions, while the ZnO was synthesized in-house using the sonochemical method [28,29], as described below.

### 2.3. Synthesis of the ZnO Nanopowder by Sonochemistry

Briefly, the sonochemical synthesis method of metal oxides is a precipitation method enhanced by the high-intensity acoustic irradiation of a liquid phase containing the dissolved precursors of the material to be fabricated. Within this physical–chemical process, there is no direct interaction of the acoustic waves with the matter at the molecular level, due to the considerable difference between the wavelength of the acoustic waves and molecule dimensions [30]. The driving force of the increased rate of the chemical processes, such as water dissociation, hydrolysis and polycondensation, is the cavitation generated by the propagation of the acoustic wave in the liquid state, characterized by the low lifetime air bubble formation. The implosion of these transient bubbles (with inner temperature and pressures of 5000 K, and 1000 bars, respectively) locally creates high temperatures and pressure waves, activating the above specific reactions.

The synthesis of ZnO powder was performed in a Hielscher Ultrasonics St apparatus (Teltow, Germany), operating at a frequency of 26 kHz; a power of 200 W was used. The volume of the liquid phase was equal to 80 mL, the longitudinally vibrating probe sonotrode immersed in the liquid phase for transmitting the acoustic waves to the liquid had a surface area of 1.53 cm^2^, and thus reaching a power density of 130 W/cm^2^.

For the synthesis of the ZnO powder, an amount of 1 g of Zn(NO_3_)_2_·6 H_2_O was dissolved in 80 mL of deionized water. The solution was added to a 100 mL recipient, to which a solution of NaOH (Sigma-Aldrich, Taufkirchen City, Germany), was added until the pH of the solution became equal to 14. Then, the sonochemical process was launched for 1.5 h. During the sonochemical synthesis, the temperature of the suspension increased to about 80 °C, and the level of suspension was kept constant by periodically adding small amounts of water. Thus, the intensity of the acoustic radiation was kept constant in the liquid phase to a value of 2.5 W/mL. At the end of the sonochemical process, the precipitate was filtered and washed a few times until the value of the pH of the supernatant solution was equal to 7. The precipitate was then dried at 150 °C and calcinated in the air at 450 °C for 1 h.

### 2.4. Synthesis of the Organic–Inorganic Ternary Nanohybrids of Ox-SWCNHs

All of the four ternary compositions of SWCNHs described above are created using similar steps. As an example, the process steps for the fabrication of the solid-state sensing films based on Ox-SWCNH/SnO_2_/PVP = 1/1/1 as mass ratios (*w*/*w*/*w*) are shown below:A solution of polyvinylpyrrolidone is prepared by dissolving 0.1 g polymer in 50 mL isopropyl alcohol under stirring in the ultrasonic bath (for one hour at a temperature of 50 °C).An amount of 0.1 g of ox-SWCNH powder is added to the previously prepared solution, while continued stirring is performed in the ultrasound bath for 2 h at room temperature.An amount of 0.1 g of SnO_2_ nanometric powder is added to the previously prepared suspension and continued stirring is performed in the ultrasound bath for 2 h at room temperature.

### 2.5. Morphological and Compositional Characterization of the Sensing Films

Surface topography of the sensing films based on ternary nanohybrids of SWCNHs was investigated by scanning electron microscopy (SEM). At the same time, the formation of the multicomponent hybrid and interaction of the carbonic material with the other organic–inorganic partners have been proven using Raman spectroscopy. For surface visualization, a field emission gun scanning electron microscope/FEG-SEM-Nova NanoSEM 630 (Thermo Scientific, Waltham, MA, USA) (FEI), with superior low voltage resolution and high surface sensitivity imaging was used. The vibrational fingerprint of the ternary organic–inorganic nanohybrids of the SWCNHs was studied directly on the films deposited on a silicon substrate. The Raman spectra have been collected at room temperature with a WITec Raman spectrometer (Alpha-SNOM 300 S, WITec. GmbH, Ulm, Germany) using 532 nm as an excitation. The 532 nm diode-pumped solid-state laser has a maximum power of 145 mW. The incident laser beam with a spot-size of about 1.0 µm was focused onto the sample with a 100 × long-working distance microscope objective. The Raman spectra were measured with an exposure time of 20 s accumulation, and the same objective collected the scattered light in back-scattering geometry with 600 grooves/mm grating. The calibration of the Raman systems was carried out using the 512 cm^−1^ Raman line of a silicon substrate, which corresponds to the longitudinal optical–transverse optical (LO–TO) phonon. The spectrometer scanning data collection and processing were carried out by a dedicated computer using WITec Suite FIVE software (WITec GmbH, Ulm, Germany).

## 3. Results and Discussion

### 3.1. Structural and Topographical Characterization of Ternary Nanohybrids of SWCNHs

The Raman spectra of the sensing films based on the above nanocomposites of the SWCNHs deposited on a silicon substrate have confirmed the formation of an organic–inorganic ternary nanohybrid, where each component contributes with its specific light scattering bands to the pattern. As an example, Figure 1 (blue color) presents the typical Raman spectra of the solid sensing film based on ox-SWCNH/SnO_2_/PVP = 1/1/1 (mass ratios of the three components), deposited on a silicon substrate. The Raman bands of the substrate (black color) and PVP (red color) are also shown separately, for easier identification of their contribution to the bands of the ternary nanocomposite.

In Figure 1, the Raman structural fingerprint of the ox-SWCNHs is seen, with the well-known first-order bands of sp^3^ defects (D), sp^2^ graphene (G), at 1336, 1558 cm^−1^, respectively, as well as second-order bands of 2D and D + D’ band localized at 2654, 2940 cm^−1^, respectively, which are in agreement with previously reported results for SWCNHs [31,32]. The broadening of the Raman bands of the SWCNHs is an indication of the mechanical stress in the carbonic material [32].

The D’ band from about 1615 cm^−1^ appears as a shoulder of the G band, but it is in general hardly evidenced by typical instruments. Additional Raman contribution from 2440 cm^−^^1^, as well as the above-mentioned D + D’ peak (~2940 cm^−^^1^) shows wide bands indicating the presence of compressed carbon nanohorns (CNHs) [32]. The Raman band from about 3217 cm^−1^ may be associated with the so-called 2D’ band and this may come from the vibration spectrum of graphite [33].

The existence of the graphite in the SWCNHs is also specified by the supplier, who mentioned its 10% concentration in the SWCNH powder. The SnO_2_ fingerprint in the Raman spectra of the ternary hybrid (blue curve) is indicated by the Raman active band from 87 cm^−1^ specific to oxygen atom vibration modes, and which is associated with optical phonons of the B1g symmetry [34]. Due to the very high intensity of the peak from 520 cm^−1^ associated with the silicon substrate [35], the Raman active bands of the SnO_2_ from 486, 568, 634, 706 cm^−1^ [36] appeared much weaker in that region of Figure 1. Additionally, as shown in Figure 1 by the red color spectrum, the PVP as a pure material has Raman spectra localized at 854, 1429, 1665, 2925, 2997 cm^−1^ [37,38], but when this is a part of the ternary nanohybrid film, most of the Raman active bands overlap with the bands of the SWCNH substrate, and even with the SnO_2_ active band below 100 cm^−1^. The Raman band from about 900 cm^−1^, which is not assigned in the spectrum of the ternary nanohybrid (blue curve), may be from the overlapping of the Raman bands of substrate and PVP. Overall, the ternary organic–inorganic nanohybrid has a complex Raman spectrum, where all the components bring their contribution to the active vibration modes of this material. Some of the small Raman shifts concerning the behavior of the pure components may also reflect the chemical interactions between the hydrogen bonds of ox-SWCNH and the hydrophilic SnO_2_ and PVP molecules.

Figure 2 shows the SEM images of ternary nanohybrid films with the compositions ox-SWCNH/SnO_2_/PVP = 1/1/1 (Figure 2a) and ox-SWCNH/SnO_2_/PVP = 2/1/1 (Figure 2b), both deposited on the silicon substrate. It appears that the surface topography of these sensing layers is determined by the SnO_2_ islands, with a maximum size of 1–2 µm obtained by the clustering of 0.1 µm SnO_2_ nanoparticles (as reported by Sigma-Aldrich supplier), surrounded by the continuous matrix of PVP and ox-SWCNHs. It may also be possible that aggregates of ox-SWCNHs and graphite, as described by Sigma-Aldrich, contribute to the surface morphology of the sensing films. A similar topography was noticed for the ox-SWCNH/ZnO/PVP sensing films. For these films, the ZnO nanopowder was obtained by sonochemistry, at a pH value of precursor suspension equal to 14 [29]. In this latter case, the ZnO powder added to the ternary composite had a nano-flower-like nanostructure, explained in our previous paper by the effect of a high concentration of hydroxyl groups on the nucleation and anisotropic growth of ZnO seeds [29].

### 3.2. Results on Chemiresistive Ethanol Detection at Room Temperature by Ternary Nanohybrids of Ox-SWCNHs

With the set-up for automatic characterization of sensors described in Appendix A, and the calibration plot of ethanol vapor concentration as a function of airflow rate passing through liquid ethanol at a constant total flow rate of 1 L/min (in the test chamber), from Appendix A, the response of the well-known interdigitated (IDT) chemiresistive test structures (from Appendix A) to ethanol vapor concentration in dry air is obtained. Thus, in Figure 3, the transfer functions of the chemiresistive IDT test structures with sensing films based on ox-SWCNH/SnO_2_/PVP = 1/1/1 (mass ratio) (Figure 3a) and ox-SWCNH/SnO_2_/PVP = 2/1/1 (mass ratio) (Figure 3b) and kept at room temperature of 21 °C are shown. From these transfer characteristics, where the sensing film is kept at room temperature equal to 21 °C, it is easy to remark that the two sensors are very sensitive to low ethanol vapor concentrations (below 1 mg/L) in the dry air of the test chamber; they preserve high sensitivities up to higher concentrations of ethanol vapors in the range of 25–50 mg/L. The limit of detection (LOD) for these ternary sensors is 0.3 mg/L. It is also evident from Figure 3 that for ethanol concentrations above 1 mg/L, both sensors have excellent linearity, too. This is a good indication of the absence of a saturation process during ethanol detection for these sensing materials. The insets of Figure 3 indicate the evolution of the sensor response as a function of measurement time and ethanol vapor concentration in the test chamber during the automatic recording process, where one measurement of electrical resistance per second was made and collected. These insets present three continuous measurement cycles, each one taking about 7000 s.

For each cycle, the ethanol vapor concentration was increased from 0 mg/L to 25 mg/L in ten steps, and then ethanol inlet was suddenly stopped in order to evaluate the recovery time of the sensor. The first cycle has been used for the fresh sensor equilibration with the ethanol-containing dry synthetic air. As can be seen in these insets, overall, the sensor response during the second and the third cycles have had enough good reversibility behavior, without any requirement to heat the sensing film in order for it to recover from the high concentration of ethanol vapors in air to clean air. In addition, the 1/1/1 sensor has proven good reusability potential, which is supported by similar behavior in the third cycle immediately following the second one, without heating in between. This also means that the sensitivity of the 1/1/1 sensor as a function of time has an enough good stability, but more studies are necessary in this direction. This is a good starting point for further sensor technology optimization. From this point of view, it appears that the 1/1/1 sensor has a lower baseline drift (Figure 3a) than the 2/1/1 sensor (Figure 3b). Therefore, this is the first hint that the 1/1/1 sensor is better performing. Figure 4 presents the transfer functions of the chemiresistive IDT test structures with sensing films based on ox-SWCNH/ZnO/PVP = 5/2/1 (mass ratio) (a) and ox-SWCNH/ZnO/PVP = 5/3/2 (mass ratio) (b). From this figure, one can note a similar sensing behavior of the organic–inorganic ternary nanocomposites based on ox-SWCNH-ZnO-PVP, with increased sensitivity at low concentrations of ethanol vapors below 1 mg/L, and a constant sensitivity up to 25 mg/L. The much higher value of the sensor resistance with the composition ox-SWCNH/ZnO/PVP = 5/3/2 may be explained by the increased amount of ZnO and PVP components and a possible local non-homogeneous distribution of ox-SWCNH. For a sensor operation with the sensing film kept at room temperature (RT), it is important to evaluate the response and recovery times, as both adsorption of gas to be detected and desorption of reaction products may be very slow under such RT circumstances. As an example, the response times and the recovery times of the chemiresistive test structures based on the two studied compositions of SWCNH/SnO_2_/PVP ternary hybrids are presented in Figure 5, where the response time is associated with the case of the ethanol vapor concentration increasing from 2 to 5 mg/L, while the recovery time corresponds to the decrease in the ethanol concentration from 25 mg/L to 0 mg/L (dry clean air).

By taking an average value of the lower and upper levels to be reached and following the standard calculations (between 0.1 and 0.9 values of the signal variation between the two levels), one obtains a response time of about 30–35 s for the test structure with the sensing layer SWCNH/SnO_2_/PVP = 1/1/1 and a value of about 15–20 s for the test structure with SWCNH/SnO_2_/PVP = 2/1/1. Similarly, the recovery time is about 90 s for the chemiresistive structure based on the sensitive layer with SWCNH/SnO_2_/PVP = 1/1/1 composition and about 50 s for the structure with SWCNH/SnO_2_/PVP = 2/1/1 composition of the sensing layer. These results, with lower values for the response times and recovery times for the devices based on SWCNH/SnO_2_/PVP = 2/1/1 composition, may be explained by the effect of the increased ox-SWCNH concentration in the film, which thus enhanced the rate of the charge transfer reactions specific to the sensing mechanism.

The comparative evaluation of these times with those reported in Figure 5 shows that higher response times (60–100 s) and recovery times (400–500 s) are obtained for the test structures based on ox-SWCNH/ZnO/PVP, under similar changes in the ethanol vapor concentration. From Figure 6, it appears again that the values of the response and recovery times are somewhat lower for the test structure with a higher ox-SWCNH concentration in the ternary hybrid. However, for both ox-SWCNH/ZnO/PVP compositions, the dynamic range of the sensor in the ethanol vapor concentration domain was rather small, and this may limit the applicability of this type of test structure.

The organic–inorganic ternary nanohybrids of this work contain ox-SWCNHs, semiconducting metal oxides such as SnO_2_ or ZnO and polymers such as PVP, which have been previously used as individual or binary composites in chemiresistive ethanol detection for alcohol detection, as will be exemplified below. However, this is the first time when all three components have been used together in a sensing film.

For a realistic evaluation of the sensing performances of the ternary organic–inorganic composites developed by our group, a short review of the present status of volatile organic compound (VOC) sensing by different principles is shown in Appendix A. Thus, it was observed that in comparison with the state of the art in VOC sensing, our ternary composite-based chemiresistive sensors exceeded the prior art performances of commercial and research sensors in terms of power consumption, response time, and recovery time.

Considering the prior state of art investigation, it appeared that this is the first time that the ethanol vapor sensing capability of the ox-SWCNHs in ternary organic–inorganic hybrids, such as ox-SWCNH/SnO_2_/PVP or ox-SWCNH/ZnO/PVP, has been studied. The experimental results from Figure 3, Figure 4, Figure 5 and Figure 6 demonstrate that low concentrations of ethanol vapors, below 1 mg/L, can be detected at room temperature, while the sensitivity remained constant for ethanol vapor concentrations increasing up to 25–50 mg/L. These results may be explained by the high specific area of the oxidized single-walled carbon nanohorns, and this is a good starting point for further development. The results also indicate that the values of the electrical resistances of the sensors at room temperature are much lower than those usually found for the metal oxides or metal oxide–polymer composites, and this fact is entirely related to the high concentrations of carbonic material in the ternary nanohybrid. More specifically, the concentration of ox-SWCNH in the studied ternary nanohybrids, which was in the range of 33–62.5 wt%, is much higher than the percolation threshold of carbonic materials in polymer matrices, which is of the order of 0.04–10 wt% [22]. Recently, we have determined the percolation threshold of ox-SWCNH in PVP polymer, and we found a value of (0.05–0.1) wt% for IDT test structures with the interelectrode gap of 10 µm [39]. The ultimate advantage of the ethanol detection at room temperature by using our developed ternary compositions was the power consumption of the studied chemiresistive sensors, which was in the range of (0.4–1.6) mW. In contrast, the commercial sensor TGS 822 requires 660 mW for heating the sensing layer, and under these conditions its response time is about 40 s while the recovery time is 300 s. The operation of our sensor much beyond the percolation threshold of the ox-SWCNH in the ternary hybrid may also explain the fast response times (15–30) s and recovery times (50–90) s obtained for the sensors based on ox-SWCNH/SnO_2_/PVP nanohybrids, even if they operate with the sensing layer kept at room temperature. In other terms, the rapid charge transfer from the ox-SWCNH-based transducer layer to the metal electrodes via the multiple percolative carbon paths is responsible for this excellent dynamic behavior. As a comparison, a chemiresistive sensor based on sensing film made of reduced graphene oxide-SnO_2_ has had a response time equal to 97 s and a recovery time of 104 s, when operated at room temperature [40].

Therefore, our 1/1/1 sensors with a low power consumption (0.4–2 mW) show shorter response and recovery times with respect to “heated” commercial sensors as well as state of the art chemiresistive sensors based on binary composites of metal oxides and carbonic materials, and these are impressive behaviors for these ternary nanohybrids-based sensors. These proven features, such as low power consumption, no need for regeneration or other auxiliary tubing, high sensitivity of the sensor at ethanol vapor concentrations below 2 mg/L in air, will create opportunities for the sensor application in both consumer applications such as alcohol testers for police portable instruments (a drunk driver can have 1 mg/L ethanol in exhaled air), as well as passive samplers for mobile wireless sensor networks and static ambient monitoring applications. With response and recovery times superior to the TGS 822 commercial sensors and a rather simple fabrication process and reading electronics, the “ternary” sensors could be transferred to industry and thus play a role in Internet of Things applications, if more development work is performed, and further optimization of the fabrication process is undertaken. For such internet applications, a rather high data collection rate of below 1 min can be achieved by these sensors, at a very low power consumption, in some cases for a few hundred of µW!

At this stage of technology development, one can notice a certain dispersion in the values of the sensor resistance prepared under “identical” conditions. These resistance variations are directly correlated with existing limitations in the uniformity of the carbon nanohorns and metal oxide nanoparticle distribution in the PVP matrix. For the comparison of sensing performances of the chemiresistive sensors, with different initial resistance values, a relative sensitivity (S_r_) to ethanol vapor concentrations (C) was defined. This S_r_ amount consists of sensitivity (S = ΔR/ΔC) of the transfer characteristics from Figure 3 and Figure 4 in the linear domain of (1–25) mg/L, divided by the initial electrical resistance of the sensor in clean dry air (R_o_). These results are presented in Table 1. It is easy to note that the sensor based on ternary hybrid Ox-SWCNH/SnO_2_/PVP = 1/1/1 has the highest relative sensitivity (S_r_), being at least two-times more sensitive than the next best alternative sensor, based on Ox-SWCNH/SnO_2_/PVP = 2/1/1. From Figure 3 and Figure 4, one can also note that the sensor with Ox-SWCNH/SnO_2_/PVP = 1/1/1 composition has the highest dynamic range from all four devices.

This comparative analysis shows that there is an optimum concentration of the ox-SWCNH, which should be used for maximizing the sensitivity of ternary organic–inorganic nanohybrids. For further optimization of this sensor in applications where higher sensitivity is a major requirement, the concentration of ox-SWCNH in the ternary nanocomposite may be varied around the value of 33 wt%, but below 50 wt%, which has already decreased the relative sensitivity for ethanol vapor detection. For the case of a possible application where the dynamic parameters of response time and recovery times are more important than the static ones, then optimization of the concentration of ox-SWCNH in the ternary nanohybrid should be achieved around Ox-SWCNH/SnO_2_/PVP = 2/1/1.

The ethanol sensing mechanism in chemiresistive test structures should consider the charge transfer reaction (1) between ethanol vapors, such as reducing gas and adsorbed ionized oxygen species. This reaction is releasing electrons to the sensing layer, following the well-known oxidation reaction: CH_3_CH_2_OH (gas) + 6O^−^ (ads) = 2CO_2_ + 3H_2_O + 6e^−^(1)

If the sensing layer is an *n*-type semiconductor, then this mobile charge that is entering the film will increase the number of conduction electrons and its electrical conductivity, while in the case of *p*-type, these electrons are recombining with the holes from the semiconductor surface accumulated layer, and thus reducing its conductivity (electrical resistance increase) [41]. As shown in Figure 3 and Figure 4, the transfer characteristics of the chemiresistive test structures indicate that the values of the electrical resistance increase as a function of ethanol vapor concentration increase, which suggests that the overall behavior of the ternary nanocomposites of ox-SWCNH/SnO_2_/PVP and ox-SWCNH/ZnO/PVP is equivalent with a *p*-type semiconducting material. This experimental result may appear rather unexpected if one considers that the semiconducting metal oxides SnO_2_ and ZnO are *n*-type, ox-SWCNH is *p*-type, while the PVP is a dielectric from an electrical conduction point of view. The fact that heterostructures built between *p*-type ox-SWCNHs and *n*-type SnO_2_ or ZnO nanoparticles of the sensing film may not play a major role in the electrical conduction and sensing process of the chemiresistive device could be explained by the presence of the percolating paths of ox-SWCNH, which shunt the multitude of formed *p-n* heterojunctions, which ultimately could influence only the geometry of the conducting avenues of the ternary hybrid. In other words, the percolative carbon paths are screening the metal oxide nanoparticles, which are thus shunted by the carbon paths, and therefore minimize the role of metal oxides and associated *p–n* heterostructures made with ox-SWCNH.

The ethanol sensing mechanism in the chemiresistive test structures based on the organic–inorganic ternary hybrids, where the ternary sensing hybrid, with an ox-SWCNH concentration higher than the percolation threshold, behaves similar to an overall *p*-type semiconductor, may be also explained from a chemical point of view in terms of the hard–soft acid–base (HSAB) principle, which states that hard acid likes hard base and soft acid likes soft base [42]. According to the HSAB principle, in the sensing process, the hydroxyl groups of ethanol, which are hard Lewis bases will react with the holes of the *p*-type sensing films that are strong Lewis acids, and therefore they will be nulling each other by their recombination process [25,26,27,43]. Thus, the number of holes as majority charge carriers in the ternary hybrid will decrease, and therefore the electrical resistance of the sensor will increase.

Further optimization of these sensors, which are able to detect ethanol from the ambient air starting from low concentrations (below 1 mg/L), may open new pathways towards the development of low power portable instruments for gas sensing in industrial and consumer applications. As an example, the alcohol test portable instruments have to detect ethanol concentrations below 1 mg/L in exhaled air, and this concentration in the air fits well with the sensing capabilities of these chemiresistive sensors based on organic–inorganic ternary hybrids of ox-SWCNHs operating at room temperature, as described above. Low power consumption (below 2 mW) may also stimulate the use of such sensors in future wireless sensor networks for Internet of Things applications, where energy constraints are limiting the use of existing commercial sensors.

## 4. Conclusions

This paper presents for the first time a room temperature ethanol chemiresistive sensor based on ternary organic–inorganic nanohybrids of oxidated carbon nanohorns, consisting of ox-SWCNH/SnO_2_/PVP or ox-SWCNH/ZnO/PVP, where the concentration of the ox-SWCNH was (33–62.5) wt%, much beyond the percolation threshold of 0.1 wt%. The comparative performance analysis has shown performance superiority of ox-SWCNH/SnO_2_/PVP = 1/1/1 (mass ratios) with respect to ox-SWCNH/ZnO/PVP.

The next challenge for the development of the technology is further improvement of the uniformity of the dispersion of the carbon nanohorns and metal oxide nanoparticles in the PVP matrix.

The “ternary” 1/1/1 sensor has shown excellent sensitivities for ethanol vapor concentrations below 2 mg/L, a very good linearity up to higher concentrations of ethanol vapor (25–50 mg/L), response time of 30 s, recovery time of 50 s, a good reversibility/reusability, and a low power consumption (0.4–2 mW). These technical features are superior to some commercial sensors, such as TGS 822 sensors, that are already present on the market. The high ethanol sensitivity of below 2 mg/L in air can be very well exploited in portable alcohol testers used by the police to measure the alcoholaemia of drivers.

These impressive performances open new pathways for existing and future VOC monitoring applications related to wireless sensor networks and Internet of Things, where data can be collected from such static samplers every minute, and still accurately follow the ethanol concentration changes in the ambient air.

## 5. Patents

The following patent applications have resulted from the work reported in this manuscript:Bogdan-Cătălin Șerban, Octavian Buiu, Cornel Cobianu, Niculae Dumbravescu, Viorel Marian Avramescu, “*Strat senzitiv ternar pentru senzor rezistiv de etanol*” (*Ternary sensitive layer for ethanol resistive sensor*), application patent, OSIM, Romania, A/00477, 31 July 2020.Bogdan-Cătălin Șerban, Octavian Buiu, Cornel Cobianu, Niculae Dumbravescu, Viorel Marian Avramescu, “*Senzor rezistiv de etanol*”, (*Ethanol resistive sensor*) application patent, OSIM, Romania, A/00480, 31 July 2020.

The intellectual property associated with these patent applications belongs to the National Institute for Research and Development in Microtechnology-IMT Bucharest.

## Figures and Tables

**Figure 1 nanomaterials-10-02552-f001:**
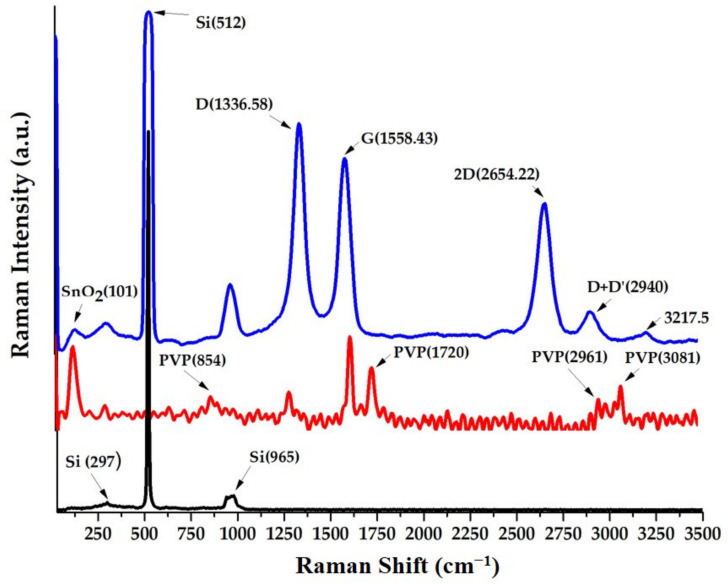
Raman spectra of solid-state films of single walled carbon nanohorns-SnO_2_-polyvinylpyrrolidone (SWCNH/SnO_2_/PVP) = 1/1/1 (mass ratio) deposited on silicon substrate.

**Figure 2 nanomaterials-10-02552-f002:**
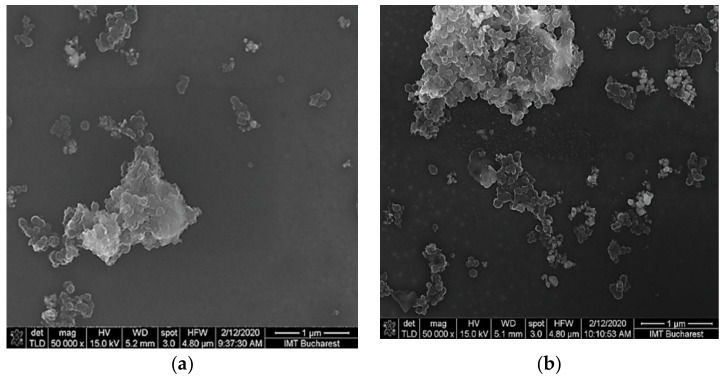
SEM images of the sensing ternary films deposited on the sensing area of the test structure. (**a**) Oxidized-single walled carbon nanohorns-SnO_2_-polyvinylpyrrolidone (ox-SWCNH/SnO_2_/PVP) = 1/1/1; (**b**) and ox-SWCNH/SnO_2_ = 2/1/1.

**Figure 3 nanomaterials-10-02552-f003:**
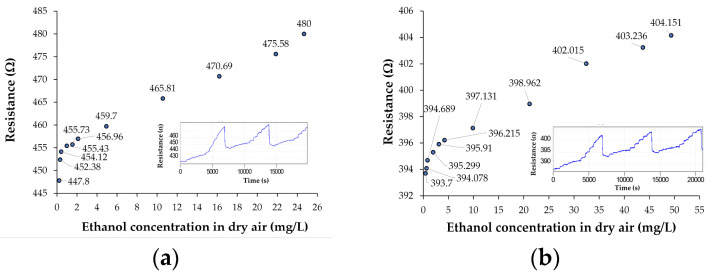
Transfer functions of the chemiresistive ethanol sensors with sensing films kept at room temperature and consisting of (**a**) ox-SWCNH/SnO_2_/PVP = 1/1/1 (mass ratio) and (**b**) ox-SWCNH/SnO_2_/PVP = 2/1/1 (mass ratio). For both cases, the flow rate of air passing through liquid ethanol was varied, while the total flow rate was kept constant at 1 L/min. The inset shows the automatic recording of the sensor resistance as a function of time and ethanol vapor concentration.

**Figure 4 nanomaterials-10-02552-f004:**
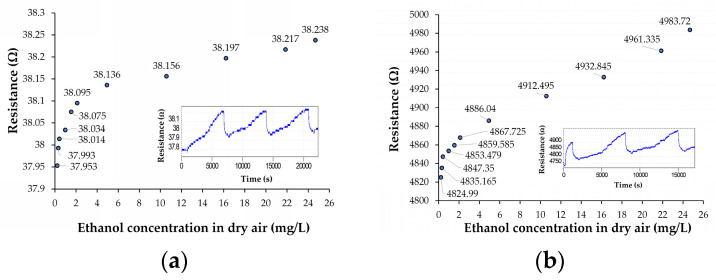
Transfer functions of the chemiresistive ethanol sensors with sensing films kept at room temperature and consisting of (**a**) ox-SWCNH/ZnO/PVP = 5/2/1 (mass ratio) and (**b**) ox-SWCNH/ZnO/PVP = 5/3/2. For both cases, the flow rate of air passing through liquid ethanol was varied, while the total flow rate was kept constant at 1 L/min. The inset shows the automatic recording of the sensor resistance as a function of time and ethanol vapor concentration.

**Figure 5 nanomaterials-10-02552-f005:**
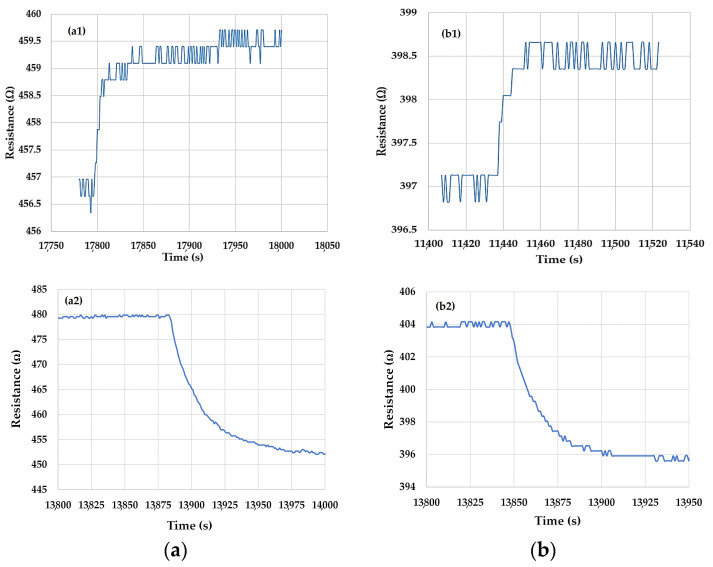
Response and the recovery times of the chemiresistive ethanol sensors with sensing films kept at room temperature and consisting of (**a**) ox-SWCNH/SnO_2_/PVP = 1/1/1 (mass ratio), where (**a1**) is response time, and (**a2**) is recovery time and (**b**) ox-SWCNH/SnO_2_/PVP = 2/1/1 (mass ratio), where (**b1**) is response time, and (**b2**) is recovery time. Response time was measured for a change in ethanol vapor concentration from 2 to 5 mg/L, while the recovery time was measured for a change in ethanol vapor concentration from 25 mg/L to 0 mg/L. For both cases, the flow rate of air passing through liquid ethanol was varied, while the total flow rate in the test chamber was kept constant at 1 L/min.

**Figure 6 nanomaterials-10-02552-f006:**
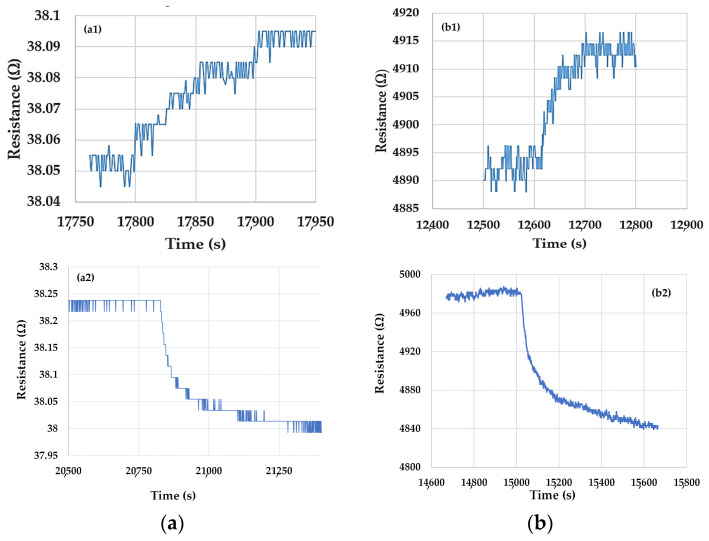
Response and recovery times of the chemiresistive ethanol sensors with sensing films kept at room temperature and consisting of (**a**) ox-SWCNH/ZnO/PVP = 5/2/1 (mass ratio); (**a1**) response time, and (**a2**) recovery time, and (**b**) ox-SWCNH/ZnO/PVP = 5/3/2 (mass ratio); (**b1**) response time, and (**b2**) recovery time. Response time was measured for a change in ethanol vapor concentration from 2 to 5 mg/L, while the recovery time was measured for a change in ethanol vapor concentration from 25 mg/L to 0 mg/L. For both cases, the flow rate of air passing through liquid ethanol was varied, while the total flow rate was kept constant at 1 L/min.

**Table 1 nanomaterials-10-02552-t001:** Comparison of sensitivities (ΔR/ΔC) and relative sensitivities (ΔR/ΔC)/R_o_ for different types of “ternary” sensors presented in this paper.

Sensor Parameters	Sensor Types			
	Ox-SWCNH/SnO_2_/PVP 1/1/1	Ox-SWCNH/SnO_2_/PVP 2/1/1	Ox-SWCNH/ZnO/PVP 5/2/1	Ox-SWCNH/ZnO/PVP 5/3/2
Resistance, R_o_ [Ω]	447.8	393.7	37.953	4824.99
Sensitivity, S = ΔR/ΔC [Ω/(mg/L)]	1.003	0.3638	0.0032	2.4213
Relative sensitivity, S_r_ = S/R_o_ [1/(mg/L)]	0.0022	0.00092	0.000084	0.0005

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
