# Peer review of "Organic–Inorganic Ternary Nanohybrids of Single-Walled Carbon Nanohorns for Room Temperature Chemiresistive Ethanol Detection"

_nanomaterials, 2020, doi:10.3390/nano10122552_

Round 1

Reviewer 1 Report

Although the authors present some interesting results on inorganic/organic hybrid ethanol sensors, the paper could be improved. In particular, the temporal response is quite long and often the reason that commercial sensors include heaters. In order to address the origin of this, the authors should present a model for the kinetics that enables the time constants for possible processes to be identified in this case. Moreover, the sensing mechanism is not well described in these complex structures and the paper would benefit from a presentation of the key mechanisms for resistive changes including charge transfer and screening, scattering etc. Finally, the sensitivity of the sensors as a function of time should be noted - ie., in terms of initial and steady state response. 

Author Response

Dear Reviewer 1,

Thank you very much for your review. It helped us to further improve our manuscript.

  1. Related to your comments on the response time and recovery time of our chemiresistive sensors saying that “the temporal response is quite long and often the reason that commercial sensors include heaters”

        Your comments from above have motivated us to extend the comparison of the dynamic performances of our sensors versus TGS 822 commercial sensor dedicated to ethanol detection. Here it is what we have found on the site of Figaro company: The response time of TGS 822 chemiresistive ethanol sensor with sensing layer heated at 350oC and consuming about 660 mW is about 40 s, while the recovery time is 300 s. The technical spec of TGS 822 is attached.

(https://cdn.sos.sk/productdata/03/a8/53fad086/tgs-822.pdf)

      Our ethanol chemiresistive sensor based on ternary nanocomposites (ox-SWCNH/SnO2/PVP) with sensing layer operating at room temperature and consuming below 2 mW have a response time in the range of (15-30) s and a recovery time of (50-90) s. Therefore, we are respectfully saying that the response and recovery times of our sensors are NOT QUITE LONG, as long as those values are smaller than those of the heated commercial sensors.

     Actually, it may not be surprisingly that, even if at room temperature, our sensors respond much faster than the commercial sensors, taking into account that our sensors operate beyond the percolation threshold of the ox-SWCNH in PVP and the electrical charges resulted from the charge transfer reactions mechanism move very rapidly from the transducer layer to the metal electrodes via the percolative conducting paths created by SWCNH. In the case of response time, it may be possible that the sensor itself to be faster than the value we measured, as long as the testing chamber needs some time to pass from a certain gas composition to another gas composition associated to the new ethanol vapor concentration. In this hypothesis, the sensor may measure “in real time” the rate of air composition change in the test chamber?  

  1. Related to your comments “in order to address the origin of this, the authors should present a model for the kinetics that enables the time constants for possible processes to be identified in this case”

     The above statement starts from the idea that our sensors have longer response and recovery times, and a model is required for explaining this “bad dynamic behavior”. In our opinion, our sensors have a “good dynamic behavior” and a kinetic model was described in the manuscript for these good results (and also above), based on rapid travel of the resulting electrical charges from transducer layer to metal electrodes via carbon percolating paths.

  1. Related to your comments that “the sensing mechanism is not well described in these complex structures and the paper would benefit from a presentation of the key mechanisms for resistive changes including charge transfer, the screening, scattering etc.)”.

    We agree with you that the paper was not entirely focused on the sensing mechanism, but you may agree that we have theoretically explained our results. Thus, in the manuscript we do describe the charge transfer reaction associated to ethanol detection in an explicit way, and, then we explain the key aspect of sensing mechanism in the ternary nanohybrid saying that ox-SWCNH as a material behaves as p-type semiconductor, and for this we cited our previous papers on binary nanohybrids like ox-SWCNH/PVP. In addition, we mention about p-n heterojunctions formed between p-type ox-SWCNH and n-type SnO2 or p-type ox-SWCNH and n-type ZnO.   Finally, we show that the percolation phenomena are “screening” the above heterojunctions, and in the end, all that matters is the p-type character of ox-SWCNH. In parallel, we explain the charge transfer reaction in terms of HSAB (Hard-Soft-Acid-Base) theory, which is an original approach, in our opinion. For sure, in the future we shall go deeper into these mechanism aspects, but for the present context, the reader will understand the key aspects of the detection mechanism and the explanation of our experimental results.

  1. Related to your comments saying that “sensitivity of the sensors as a function of time should be noted-ie., in terms of initial and steady state response”.

Thank you for the comment. Indeed, this is the reason why we have inserted those insets in the figures 3-4, where we wanted to see how the sensitivities of the sensors varied as a function of time during those measurement cycles, for different values of the ethanol vapor concentrations. Some sensors perform better than others from this point of view. It appears that 1/1/1 Ox-SWCNH/SnO2/PVP sensors are better preserving the sensitivity during different periods of time. The key aspect here is also the baseline stability in time, and from this point of view the 1/1/1 Ox-SWCNH/SnO2/PVP sensors appear to behave better, too.  But we do agree with you that this is an early stage of the technology development and more work will be needed before robust stabilization of the characteristics is obtained.       

    Finally, we would like to thank you again for your advanced comments and time allocated to the review of our manuscript.

Kind Regards,

The authors   

Reviewer 2 Report

I would suggest major revision, as detailed below:

1. Title. The authors should specify the technique of detection.
2. Introduction. The text contains trivial statements (for instance the first three paragraphs). The introduction should be reduced in length and have a focus on current technological and analytical challenges.
3. Experimental. Sub-sections 2.5, 2.6 (inadvertently reported 2.5) and 2.8 contain so many technical details that make reader feel bored. They should be shifted to a Supplementary material.
4. We use to define sensitivity as the slope of a calibration curve (roughly speaking). As the results are given here, some figures (e.g.
0.0022 (mg/L-1)) represent the quantitation (or detection) limit.
5. The manuscript is very verbose. Results and discussion should merge and be shortened in order for the readers to be more concentrated on the useful results.
6. Reversibility of response: How is the reversibility of the "sensor"? No data are given. Can the sensor be reused? Is a specific regeneration protocol needed to be applied?
7. It is not clear how the authors conceive the application of this chemiresistive prototype. Can it be used under static conditions, as passive samplers? Analytical figures of merit are missing.
8. The superior performance of the material used has to be demonstrated more impressively. It should be compared to previous methods discussing various features. The work has to show a significant application of the material.
9. A conclusion is not an abstract. Focus on conclusions, on the scope (such as applicability to related detection schemes), and also on limitations, but do not repeat the abstract.

Author Response

Dear Reviewer 2

We are greatly appreciating your comments for further improving our manuscript.

Below we try to give answers in Italics to your questions and comments as follows:

Reviewer 2 : I would suggest major revision, as detailed below:

  1. The authors should specify the technique of detection.

Authors’ answer: Thank you.! We have changed the title of the manuscript, by adding the “chemiresistive” term to the initial title. The new title is: Inorganic-Organic Ternary Nanohybrids of Single-Walled Carbon Nanohorns for Room Temperature Chemiresistive Ethanol Detection  

  1. The text contains trivial statements (for instance the first three paragraphs). The introduction should be reduced in length and have a focus on current technological and analytical challenges.

Authors’ answer: We have removed the first three “trivial” paragraphs and replaced them  by a single and condensed paragraph.  

  1. Experimental. Sub-sections 2.5, 2.6 (inadvertently reported 2.5) and 2.8 contain so many technical details that make reader feel bored. They should be shifted to a Supplementary material.

          Authors’ answer: We have created a Supplementary material and shifted the recommended sub-sections in that supplementary document.

  1. We use to define sensitivity as the slope of a calibration curve (roughly speaking). As the results are given here, some figures (e.g. 0.0022 (mg/L-1)) represent the quantitation (or detection) limit.

   Authors’ answer: Indeed, we do have sensitivity defined and measured as the slope of the transfer functions in Fig. 3 of our manuscript. In this case the sensitivity (ΔR/ΔC) is measured in /(mg/L).  For sensor comparison, we have considered only linear region of the transfer functions.  In order to be able to compare sensors with different values of the (initial) resistance in clean air, we have defined also a relative sensitivity (Sr) as Sr=(ΔR/ΔC)/Ro.  The units for the Sr are [Sr]=[1/(mg/L)]=[L/mg]=L * mg-1.   Therefore, in our Table 1, the relative sensitivity, Sr, is measured in [L/mg]. Unfortunately, you have a mistake in your comments from above saying that the relative sensitivity from Table 1 is measured in mg/L-1=mg * L. We hope you reconsider your statement #4.   

  1. The manuscript is very verbose. Results and discussion should merge and be shortened in order for the readers to be more concentrated on the useful results.

Authors’ answer:  Indeed, we agree that the manuscript is “verbose”, but we tried to be very clear for the reader, and even for non-expert readers, as per IEEE requirement from our younger times. We minimized the number of words in different places.  In agreement with your observation, we have also shifted the prior art description of the ethanol sensors to the Supplementary material, and thus the reader to stay focused on our own results. Hopefully, you accept this.

  1. Reversibility of response: How is the reversibility of the "sensor"? No data are given. Can the sensor be reused? Is a specific regeneration protocol needed to be applied?

Authors’ answer: Thank you for the comment. Indeed, based on your observation, in the revised manuscript we have made explicit comments about the reversibility of our sensors taking into consideration insets from Figs. 3 and 4 where the three measurement cycles were following one after the other without any heating of the sensing layer in between those cycles. Therefore, the present results, from room temperature are obtained without any specific regeneration protocol. There is still some baseline drift, which may affect the reversibility of the sensor, but the 1/1/1 sensor perform enough well from this point of view. As mentioned in the manuscript, the first cycle may not be taken in the consideration, as it was a kind sensor equilibration in the ambient to be detected, but at room temperature. Anyway, we are still in the research stage and such reversibility issue will be the major challenge for future research and development.       

  1. It is not clear how the authors conceive the application of this chemiresistive prototype. Can it be used under static conditions, as passive samplers? Analytical figures of merit are missing.

Authors’ answer:  Thank you for the comment. This type of sensors is conceived for ethanol sensing in low power applications. The way we tested the sensor and the results obtained, with good sensitivities for concentrations of ethanol in air below 2 mg/L may recommend this sensor for portable alcohol testers. In a larger sense, the early-stage testing show that this type of sensors operates without regeneration or other auxiliary means like tubing. Therefore, it could be a very good passive sampler for ethanol monitoring in ambient air, or in different industrial spaces with ethanol vapor emanations in the air. At this stage, we are in the position to mention the limit of detection which is about 0.3 mg/L. The repeatability of the sensor measurement has been checked by the multiple measurement cycling along 20000 s, and some baseline drift were identified. The thing is that 111 sensors have had a better behavior from this point of view. Now, we are running novel experimental measurements to evaluate the sensitivity of these sensors to the relative humidity. So, more work to be done in the future on selectivity and reproducibility, so that in the end to be able to make an exhaustive analysis of figures of merit of these sensors.      

  1.         The superior performance of the material used has to be demonstrated more impressively. It should be compared to previous methods discussing various features. The work has to show a significant application of the material.

Authors’ answer: Thank you for the comment. Based on your comments we have given “more color” to our revised manuscript and we have better emphasized the features and behavior of our ethanol sensor. In the initial manuscript we have presented a brief state of the art in this field based on the relevant research in ethanol sensors, their technology and performances underlining the operation temperature and the response time evolution as a function of sensing layers composition. Based on your idea of using Supplementary materials, this portion of text related to “previous methods and features” is now a sub-section. Hope you agree with this change of the manuscript.    

  1. A conclusion is not an abstract. Focus on conclusions, on the scope (such as applicability to related detection schemes), and also on limitations, but do not repeat the abstract.

Authors’ answer: Thank for your observation. The conclusion was entirely revised accordingly.

Overall, a BIG THANK YOU  to you dear Reviewer 2 for all your valuable comments, which contributed to the improvement of our manuscript and better oriented us to emerging applications! 

Reviewer 3 Report

This is an excellent work. It is very important in the development of ethanol sensors that are capable of operating at room temperature. This has potential advantages in portable devices applications.

The paper is well written. The only concern I have is that Figure 1 looks blurry except the label "Au/Cr electrodes and digits". I suggest that you improve the quality of Figure 1. The drawings need to be of better quality and the labels should be clear"

Author Response

Dear Reviewer 3,
Thank you so much for your comments, which will further encourage the team to work on this direction. 

We have redrawn the figures so that to improve their quality.  

Round 2

Reviewer 2 Report

The authors have taken into consideration the point raised by the reviewing.

A minor spell check is required.